# Mume Fructus reduces interleukin-1 beta-induced cartilage degradation via MAPK downregulation in rat articular chondrocytes

**Doo Ri Park**[1]☯, **Bo Ram Choi**[1]☯, **Changhwan Yeo**[1], **Jee Eun Yoon**[1], **Eun Young Hong**[1], **Seung Ho Baek**[2], **Yoon Jae Lee**[1], **In-Hyuk Ha**[1]*

1 Jaseng Spine and Joint Research Institute, Jaseng Medical Foundation, Gangnam-gu, Seoul, Republic of Korea, 2 College of Korean Medicine, Dongguk University, Goyang, Gyeonggi Province, Republic of Korea

☯ These authors contributed equally to this work.
* hanihata@gmail.com

**Data Availability Statement:** All relevant data are within the manuscript and its Supporting information files.

## Abstract

Osteoarthritis is the most prevalent type of degenerative arthritis. It is characterized by persistent pain, joint dysfunction, and physical disability. Pain relief and inflammation control are prioritised during osteoarthritis treatment Mume Fructus (Omae), a fumigated product of the *Prunus mume* fruit, is used as a traditional medicine in several Asian countries. However, its therapeutic mechanism of action and effects on osteoarthritis and articular chondrocytes remain unknown. In this study, we analyzed the anti-osteoarthritis and articular regenerative effects of Mume Fructus extract on rat chondrocytes. Mume Fructus treatment reduced the interleukin-1β-induced expression of matrix metalloproteinase 3, matrix metalloproteinase 13, and a disintegrin and metalloproteinase with thrombospondin type 1 motifs 5. Additionally, it enhanced collagen type II alpha 1 chain and aggrecan accumulation in rat chondrocytes. Furthermore, Mume Fructus treatment regulated the inflammatory cytokine levels, mitogen-activated protein kinase phosphorylation, and nuclear factor-kappa B activation. Overall, our results demonstrated that Mume Fructus inhibits osteoarthritis progression by inhibiting the nuclear factor-kappa B and mitogen-activated protein kinase pathways to reduce the levels of inflammatory cytokines and prevent cartilage degeneration. Therefore, Mume Fructus may be a potential therapeutic option for osteoarthritis.

## Introduction

Osteoarthritis (OA) is the most prevalent type of degenerative joint disorder, characterized by cartilage degradation, osteophyte formation, and synovial inflammation [1–3]. Articular cartilage primarily consists of the extracellular matrix (ECM) and chondrocytes. In chondrocytes, the activation of biochemical pathways leads to OA-causing mechanisms, which disrupt the production of cartilage-specific ECM components [4,5]. Activation of the biochemical pathway activation involves pro-inflammatory cytokine production, inflammation, and ECM degradation by catabolic matrix-degrading enzymes, such as matrix metalloproteinases (MMPs) and A disintegrin and metalloproteinases with thrombospondin motifs (ADAMTS) [6–8].

**Funding:** This work was supported by the Jaseng Medical Foundation, Republic of Korea. The funders had no role in study design, data collection and analysis, decision to publish, or preparation of the manuscript.

**Competing interests:** The authors have declared that no competing interests exist.

Upon the activation of catabolic enzymes, proteoglycans and collagen in the articular cartilage are degraded [7].

Various cytokines and enzymes, such as interleukin (IL)-1β and collagenase, cause articular cartilage degeneration; specifically, IL-1β is a major inducer of OA [9–11]. The effects of IL-1β on chondrocytes include the upregulation of MMP1, MMP3, MMP13, and aggrecanase 1 and 2 (ADAMTS4 and ADAMTS5) activity; induction of pro-inflammatory cytokine expression; and destruction of collagen type II alpha 1 chain (COL2A1) [12–15]. Furthermore, IL-1β can increase the levels of nitric oxide (NO) in the articular cavity, which is a representative pro-inflammatory mediator, similar to IL-6, tumour necrosis factor-alpha (TNF-α), and cyclooxygenase-2 (COX-2) [16]. During this process of cartilage destruction, IL-1β can activate the signaling of all three mitogen-activated protein kinases (MAPKs), extracellular-signal-regulated kinase (ERK), p38, and Jun N-terminal kinase (JNK), and that of nuclear factor-kappa B (NF-κB) [17–19].

According to the Chinese Pharmacopoeia (Chinese-Pharmacopoeia-Commission, 2020), Mume Fructus (MF, commonly known as Omae), a traditional drug and health food, is processed from the *Prunus mume* (PM) fruit (commonly known as Measil) by fumigating it at a low temperature (40°C) until it turns black. MF has been traditionally used to treat cough, dysentery, asthenia, consumptive thirst, vomiting, and abdominal pain. Recently, it has also been reported to exhibit antibacterial, antioxidant, antitussive, antiallergic, antiviral, antitumour, and anti-inflammatory effects [20–26]. Furthermore, MF is the main herbal ingredient in 'Jaseng Woongayoungsin-hwan', which has been clinically used to treat herniated discs and lower limb pain. Woongayoungsin-hwan is an extract purified from a mixture of oriental herbs (*Atractylodis* rhizoma alba, *Crataegus pinnatifida*, *Pueraria lobata* Ohwi, *Cyperus rotundus* L., *Chaenomeles sinensis* Koehne, *Panax ginseng* C.A. Meyer, *Alpiniae officinarum* rhizoma, *Cinnamomum zeylanicum* Breyne, *Triticum aestivum* L., *Zingiber officinale* Roscoe, *Citrus unshiu* Markovich, *Amomum villosum* Lour., *Piper longum* L., *Lemmaphyllum microphyllum*, *Syzygium aromaticum*, *Inula helenium*, *Pterocarpus indicus*, *Mentha arvensis* var. *piperascens*, *Arisaema amurense*, and Mume Fructus) [27]. For patients with a herniated intervertebral disc, herbal medicine treatment with Woongayoungsin-hwan has been proven effective in reducing low back pain, showing an improvement with a quick return to the activities of daily life activities and routine [27]. For patients with muscular atrophy in the lower limbs caused by a condition of peripheral neuropathy, Korean medicine treatment with Woongayoungsin-hwan has exhibited a significant therapeutic effect with a gradual increase in the duration of self-walking exercise time [28]. Disc pain and osteoarthritis have a similar etiopathogenesis, since proteoglycans, predominantly aggrecan, in the ECM of the nucleus pulposus and articular cartilage are lost in both cases [29]. However, the effects of MF on articular cartilage chondrocytes and OA progression remain unclear.

Considering the anti-inflammatory effect of MF and analgesic effect of Woongayoungsin-hwan on spinal disc problems, we hypothesized that MF is also effective against osteoarthritis. Therefore, in this study, we aimed to investigate the effects of MF extracts on IL-1β-induced OA progression in rat chondrocytes and to elucidate the potential underlying mechanisms.

## Materials and methods

### Preparation of MF and PM extracts

MF and PM were purchased from Green M. P. Pharm. Co., Ltd. (Gyeonggi-do, South Korea). PM was purchased as a dried powder. The aqueous extract of MF was prepared with distilled water, boiled using a reflux apparatus maintained at 88°C for 6 h, cooled to 20–22°C, and filtered using filter paper. The filtrate was cooled to −20°C and lyophilized using a freeze dryer

(Ilshin BioBase Co., Ltd., Gyeonggi-do, Korea) to obtain a dry MF extract, which was stored at −20˚C until further use.

## Ethics statement

All experiments involving animals were performed in accordance with the Institutional Animal Care and Use Committee (Protocol No: JSR-2021-06-002-A) of the Jaseng Spine and Joint Research Institute and they conformed to the National Research Council Guidelines to minimize suffering.

## Primary culture of rat articular chondrocytes

The 3–4-day-old postnatal Sprague—Dawley rats (Orient Bio, Seongnam, Korea) were anesthetized with 2–3% isoflurane gas (Forane; BK Pham, Goyang, Korea). Articular tissues of the animals were isolated from femoral condyles and tibial plateaus. Then, the primary chondrocytes were obtained by digestion with 0.2% collagenase type II, followed by solubilisation in serum-free Dulbecco's modified Eagle's medium (DMEM; HyClone, Logan, UT, USA) at 37˚C in a 5% CO2 atmosphere [30]. The cells were maintained in DMEM containing 1% penicillin/streptomycin (Gibco, Grand Island, NY, USA) and 10% fetal bovine serum (FBS, HyClone). Cells were treated with various doses of MF or PM and with 10 ng/mL IL-1β for 30 h.

## RAW 264.7 cell culture

The murine macrophage RAW 264.7 cell line was purchased from the American Type Culture Collection (Manassas, VA, USA) and maintained in DMEM, supplemented with 1% penicillin/streptomycin and 10% FBS at 37˚C.

## Cell viability

Primary chondrocytes ($1 \times 10^4$ cells/well) were seeded into 96-well culture plates and cultured for 2 days. The cells were treated with various concentrations of MF or PM and with 10 ng/mL IL-1β. RAW 264.7 cells ($1.5 \times 10^4$ cells/well) were seeded in 96-well plates for 16 h, incubated with various doses of MF, and treated with 1 μg/mL lipopolysaccharide (LPS; Sigma-Aldrich, St Louis, MO, USA). After 24 h, the Cell Counting Kit-8 (CCK-8; Dojin-do, Kumamoto, Japan) solution was added to each well, and the plate was incubated for 4 h at 37˚C. A microplate reader (EpochTM; BioTek, Winooski, VT, USA) was used to assess the cell viability at 450 nm.

## Nitric oxide (NO) assay

The RAW 264.7 cells were seeded at a density of $1.5 \times 10^4$ cells/well in 96-well plates, incubated at 37˚C under 5% $CO_2$ for 16 h, and treated with different MF concentrations. After 1 h of treatment, the cells were stimulated with 1 μg/mL LPS for 24 h.

The NO assay reagent was prepared prior to experiments. For preparing reagent A, 1 g of sulfanilamide (Sigma-Aldrich) and 85% phosphoric acid (Junsei Honsha Co., Ltd., Tokyo, Japan) were dissolved in 50 mL distilled water. For preparing reagent B, 1 g of N-(1-naphthyl) ethylene diamine (Sigma-Aldrich) was dissolved in 50 mL distilled water. Next, 50 μL of the culture medium and an equal volume of NO assay reagent were added to a 96-well microplate in the order A and B. The absorbance was measured at 560 nm using a microplate reader (Bio-Tek). The amount of nitrite in the medium was calculated using the sodium nitrite standard curve.

## RNA isolation and quantitative reverse transcription PCR (qRT-PCR)

Primary chondrocytes ($5 \times 10^5$ cells/well) were seeded in a 6-well culture plates. After incubation for 2 days, the cells were treated with various doses of MF or PM and 10 ng/mL IL-1β at 37˚C for 30 h. RAW 264.7 cells ($1 \times 10^6$ cells/well) were seeded in 6-well plates for 16 h, incubated with various concentrations of MF, and treated with LPS (1 μg/mL). Total RNA was isolated from chondrocytes using TRIzol™ reagent (Life Technologies, Carlsbad, CA, USA). The RNA samples were reverse-transcribed to complementary DNA (cDNA) using an RT-Kit™ (Biofact, Daejeon, Korea) following the manufacturer's instructions. qRT-PCR analysis was performed using SYBR® Green Master mix (Bio-Rad, Hercules, CA, USA) and an iCycler iQ™ real-time PCR detection system (Bio-Rad). Results were normalized using *β-actin* as an internal control. The primer sequences are listed in Table 1.

## Western blotting and immunoprecipitation

Primary chondrocytes ($5 \times 10^5$/well) were seeded into a 6-well culture plates. After incubation for 2 days, the cells were treated with different MF concentrations and 10 ng/mL IL-1β at 37˚C for 30 h. Chondrocytes were lysed using radioimmunoprecipitation assay lysis buffer (Biosesang, Sungnam, Korea) containing protease and phosphatase inhibitors. The primary antibodies used are listed in Table 2. Anti-β-actin (Santa Cruz Biotechnologies, Santa Cruz, CA, USA) and anti-glyceraldehyde-3-phosphate dehydrogenase (GAPDH) antibodies (Santa Cruz Biotechnologies) were used as the loading controls. The protein bands were visualized using an Amersham Imager 600 imaging system (GE Healthcare Life Sciences, Uppsala, Sweden) and an enhanced chemiluminescence (ECL) system (Bio-Rad, Hercules, CA, USA). The protein levels were quantified using ImageJ (NIH, Bethesda, Maryland, USA).

## Immunofluorescence assay

Primary chondrocytes were seeded onto glass coverslips in a 24-well culture plates. After incubation at 37˚C for 2 days, the cells were treated with MF at different concentrations and 10 ng/mL IL-1β at 37˚C for 30 h. Subsequently, the cells were fixed with 4% paraformaldehyde and incubated overnight with the primary antibodies (Table 3) at 4˚C overnight. After washing with phosphate-buffered saline (PBS), the chondrocytes were incubated with fluorescein isothiocyanate (FITC)-conjugated secondary antibodies (Jackson Immuno-Research Labs, West Grove, PA, USA) for 2 h at 20–25˚C. The nuclei were stained with 4',6-diamidino-2-phenylindole, dihydrochloride (DAPI) (1 μg/mL in PBS) for 10 min. Images of the cells were obtained using a confocal microscope (Eclipse C2 Plus; Nikon, Konan, Minato-ku, Japan). The fluorescence intensity was analyzed using ImageJ (NIH, Bethesda, Maryland, USA).

## MF and PM preparation for high-performance liquid chromatography (HPLC)

Powdered dried samples (1 g) were mixed with methanol:water (7:3; v/v, 10 mL) and extracted via sonication at 24–26˚C for 60 min. The extract was filtered using a 0.45 μm syringe filter, and the crude extract obtained was analyzed directly using high-performance liquid chromatography with diode-array detection (HPLC-DAD). The PM and MF filtrates were subjected to HPLC for quality control. Neochlorogenic acid, chlorogenic acid, and 4-CQA (catalogue #BP0083, #BP0345, and #BP0411, respectively; Chengdu Biopurify Phytochemicals Ltd., Chengdu, China) were used as standard materials. HPLC was performed using an 1260 Infinity™ HPLC system (Agilent Technologies, Waldbronn, Germany) equipped with a quaternary pump, DAD, auto-sampler, thermostatically controlled column compartment, and a Capcell

**Table 1. Oligonucleotides used for quantitative reverse transcription-PCR analysis.**

| Gene | Strand | Primer sequence | Origin |
|---|---|---|---|
| MMP3 | Sense<br>Antisense | 5′-ATGATGAACGATGGACAGATGA-3′<br>5′-CATTGGCTGAGTGAAAGAGACC-3′ | Rat |
| MMP13 | Sense<br>Antisense | 5′-TGCTGCATACGAGCATCCAT-3′<br>5′-TGTCCTCAAAGTGAACCGCA-3′ | Rat |
| ADAMTS5 | Sense<br>Antisense | 5′-CAAGTGTGGAGTGTGTGGAG-3′<br>5′-GTCTTTGGCTTTGAACTGTCG-3′ | Rat |
| COL2A1 | Sense<br>Antisense | 5′-TCAACAATGGGAAGGCGTGAG-3′<br>5′-GTTCACGTACACTGCCCTGAAG-3′ | Rat |
| ACAN | Sense<br>Antisense | 5′-GCCTCTCAAGCCCTTGTCTG-3′<br>5′-GATCTCACACAGGTCCCCTC-3′ | Rat |
| TNF-*α* | Sense<br>Antisense | 5′-CCGACTACGTGCTCCTCACC-3′<br>5′-CTCCAAAGTAGACCTGCCCG-3′ | Rat |
| IL-6 | Sense<br>Antisense | 5′-CCACCCACAACAGACCAGTA-3′<br>5′-GGAACTCCAGAAGACCAGAGC-3′ | Rat |
| IL-1β | Sense<br>Antisense | 5′-TTGCTTCCAAGCCCTTGACT-3′<br>5′-GGTCGTCATCATCCCACGAG-3′ | Rat |
| *β*-ACTIN | Sense<br>Antisense | 5′-GCTACAGCTTCACCACCACA-3′<br>5′-GCCATCTCTTGCTCGAAGTC-3′ | Rat |
| TNF-*α* | Sense<br>Antisense | 5′-TCCCAGGTTCTCTTCAAGGGA-3′<br>5′-GGTGAGGAGCACGTAGTCGG-3′ | *Mus musculus* |
| IL-6 | Sense<br>Antisense | 5′-GAGGATACCACTCCCAACAGACC-3′<br>5′-AAGTGCATCATCGTTGTTCATACA-3′ | *Mus musculus* |
| IL-1β | Sense<br>Antisense | 5′-CACAGCAGCACATCAACAAG-3′<br>5′-GTGCTCATGTCCTCATCCTG-3′ | *Mus musculus* |
| *β*-ACTIN | Sense<br>Antisense | 5′-TGGAATCCTGTGGCATCCATGAAAC-3′<br>5′-TAAAACGCAGCTCAGTAACAGTCCG-3′ | *Mus musculus* |

MMP3, metalloproteinase-3; MMP13, metalloproteinase-13; ADAMTS5, A disintegrin and metalloproteinase with thrombospondin motifs 5; COL2A1, collagen type II alpha 1 chain; ACAN, aggrecan; TNF-*α*, tumor necrosis factor-alpha; IL-6, interleukin 6; IL-1β, interleukin 1 beta.

pak$^®$ AQ C18 column (250 mm × 4.6 mm, 5 μm). The column temperature was set to 40˚C. The mobile phase consisting of 1% acetic acid (solvent A), and acetonitrile (Solvent B) was used for isocratic elution; the latter was conducted with 90–75% solvent A for 1–25 min. The flow rate and injection volume were 0.7 mL/min and 10 μL, respectively. Absorbance was monitored at 328 nm using DAD spectrophotometer. Compound structures were drawn using Chemdraw Ultra 7.0 (BioByte Corp., Claremont, CA, USA).

## Quantification and statistical analysis

Data were analyzed using Prism 8 software (GraphPad). All data are expressed as the mean ± standard deviation (SD) values from at least three independent experiments. Statistical analyses were performed using a one-way analysis of variance (ANOVA) to determine the differences between groups. Dunnett's multiple comparison test was performed to determine significant differences between the groups. Statistical significance was set at $P < 0.05$.

## Results

### Effect of *Mume Fructus* (MF) and *Prunus mume* (PM) on IL-1β-induced OA signaling in rat articular chondrocytes

MF is processed from PM fruits via fumigation. Therefore, the effects of PM and MF were compared to confirm the effect of PM on the primary chondrocytes in rats. The cytotoxic

**Table 2. Information of antibodies used in western blotting.**

| Antibody | Company | Dilution | Product no. |
|---|---|---|---|
| β-Actin | Santa Cruz | 1:1000 | sc-47778 |
| MMP3 | Abcam | 1:1000 | ab53015 |
| MMP13 | Abcam | 1:1000 | ab39012 |
| ADAMTS5 | Abcam | 1:500 | ab41037 |
| COL2A1 | Santa Cruz | 1:1000 | sc-52658 |
| NF-κB p65 | Cell signaling | 1:1000 | 8242s |
| p-NF-κB p65 | Cell signaling | 1:1000 | #3033 |
| ERK | Santa Cruz | 1:1000 | sc-81457 |
| p-ERK | Cell signaling | 1:2000 | #4370 |
| JNK | Cell signaling | 1:1000 | #9252 |
| p-JNK | Cell signaling | 1:2000 | #9255 |
| P38 | Cell signaling | 1:1000 | #8690 |
| p-P38 | Cell signaling | 1:1000 | #4511 |
| COX-2 | Cell signaling | 1:1000 | #12282 |

MMP3, metalloproteinase-3; MMP13, metalloproteinase-13; ADAMTS5, A disintegrin and metalloproteinase with thrombospondin motifs 5; COL2A1, collagen type II alpha 1 chain; NFκB, nuclear factor kappa B; p-NFκB, phohphorylated nuclear factor kappa B; ERK, extracellular signal-regulated kinase; p-ERK, phohphorylated extracellular signal-regulated kinase; JNK, c-Jun N-terminal kinase; p-JNK, phohphorylated c-Jun N-terminal kinase; p-P38, phohphorylated P38; COX-2, cyclooxygenase-2.

effects of PM on rat chondrocytes were evaluated using a CCK-8 assay. The viability of PM- or IL-1β-treated chondrocytes remained unaffected at concentrations ranging from 0 to 100 μg/mL (Fig 1A). Compared with MF, PM did not affect IL-1β-induced OA signaling in the primary chondrocytes of rats (Fig 1B and 1C). These results were confirmed using immunofluorescence staining (Fig 2). MF has a more significant therapeutic effect on chondrocytes than PM.

## HPLC analysis identified the chemical composition of Mume Fructus (MF) and *Prunus mume* (PM) aqueous extracts

To determine the differences in the chemical composition of MF and PM, which could be linked to their diverse effects on rat primary chondrocytes, phytochemical screening and analysis were performed using HPLC (Table 4).

HPLC chromatograms showed peaks of neochlorogenic acid (Fig 3A), chlorogenic acid (Fig 3B), and cryptochlorogenic acid (4-CQA, Fig 3C) in both MF and PM extracts. Unlike the peak area distribution of the PM extract, the peak area of 4-CQA in the MF extract was more than 4 times larger than that in the PM extract; the peak area ratio in the MF extract was

**Table 3. Information of antibodies used in the immunofluorescence assay.**

| Antibody | Company | Dilution | Product no. |
|---|---|---|---|
| MMP 3 | Proteintech | 1:200 | 17873-1-ap |
| ADAMTS5 | Abcam | 1:100 | ab41037 |
| COL2A1 | Proteintech | 1:100 | 15943-1-ap |

MMP3, metalloproteinase-3; ADAMTS5, A disintegrin and metalloproteinase with thrombospondin motifs 5; COL2A1, collagen type II alpha 1 chain.

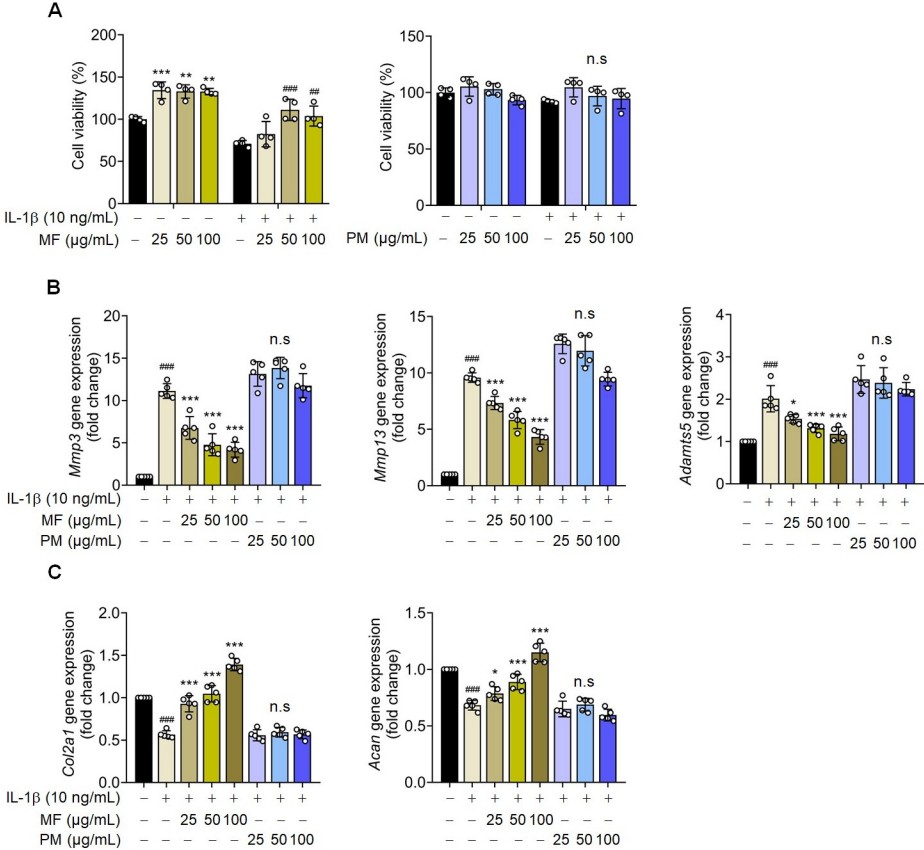

**Fig 1. Comparison of the cytotoxicity and chondrogenic gene expression efficacy of Mume Fructus (MF) and *Prunus mume* (PM) on primary rat chondrocytes.** Rat articular chondrocytes were treated with or without MF and PM extracts (25, 50, and 100 μg/mL) and exposed to IL-1β (10 ng/ml) for 30 h. (A) Cell viability was assessed using the CCK-8 assay (n = 4). (B, C) Metalloproteinase-3 (MMP3), metalloproteinase-13 (MMP13), a disintegrin and metalloproteinase with thrombospondin motifs 5 (ADAMTS5), collagen type II alpha 1 chain (COL2A1), and aggrecan (ACAN) mRNA levels were assessed using quantitative reverse transcription-PCR (n = 5). Data are expressed as mean ± standard deviation (SD). One-way ANOVA was performed, followed by Dunnett's multiple comparison test. *P < 0.05, **P < 0.01, ***P < 0.001 vs. IL-1β-treated group. #P < 0.05, ##P < 0.01, ###P < 0.001 vs. control group.

22.62% whereas that in the PM extract was 5.42% (Table 5), indicating a marked difference between the two peak areas. These results suggest that the 4-CQA component in MF inhibits OA signaling in the primary chondrocytes of rats. These results suggest that the 4-CQA component in MF inhibits OA signaling in the primary chondrocytes of rats.

## Mume Fructus (MF) treatment suppresses IL-1β-induced OA signaling in rat articular chondrocytes

The CCK-8 assay was used to confirm the cytotoxic effects of MF on rat chondrocytes. The viability of MF-treated rat chondrocytes remained unaffected at MF concentrations ranging from 0 to 100 μg/mL, and the survival rate was dose-dependent in IL-1β-treated rat chondrocytes (Fig 1A). IL-1β is known to trigger the expression of various extracellular proteolytic enzymes in chondrocytes, such as *MMP3*, *MMP13*, and *ADAMTS5* [13,14]. To elucidate the effect of MF extract on ECM synthesis, we examined IL-1β-induced ECM degradation in rat articular

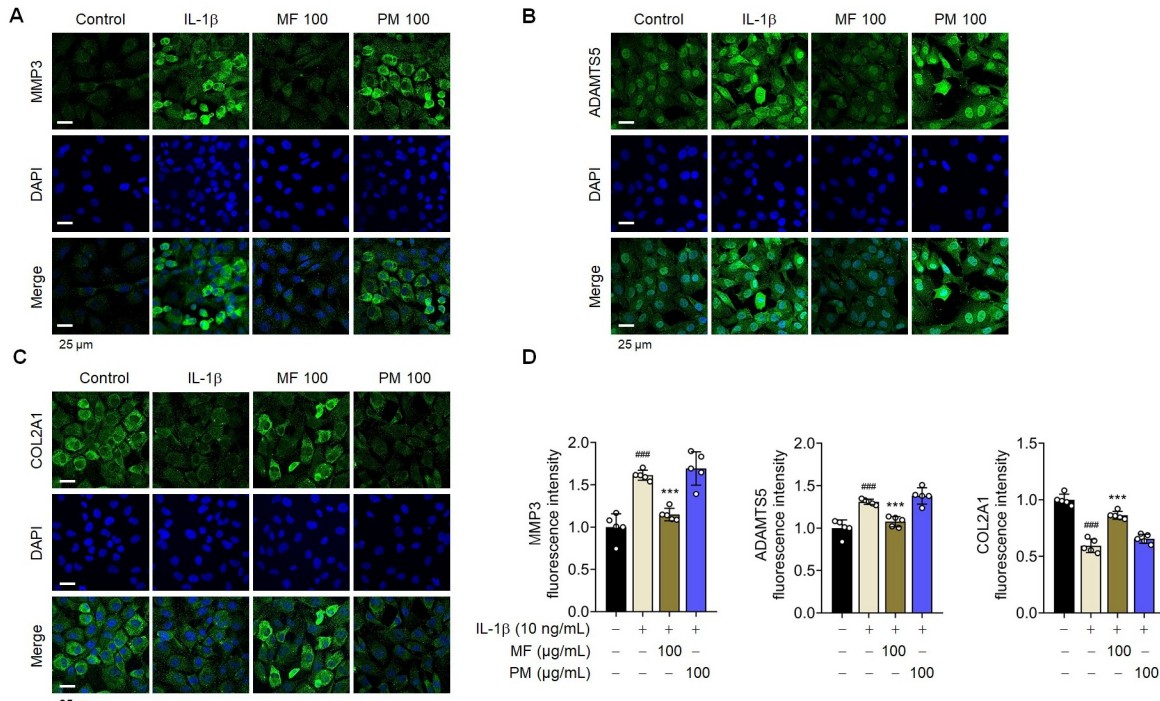

**Fig 2. *Prunus mume* (PM) had no effect on IL-1β-induced signaling in primary chondrocytes.** Expression of metalloproteinase-3 (MMP3) (A), a disintegrin and metalloproteinase with thrombospondin motifs 5 (ADAMTS5) (B), and collagen type II alpha 1 chain (COL2A1) (C) was evaluated using immunofluorescence staining (n = 5). Scale bar = 25 μm. (D) The fluorescence intensity was quantified using Image J. Data are expressed as mean ± standard deviation (SD). One-way ANOVA was performed, followed by Dunnett's multiple comparison test. *$P < 0.05$, **$P < 0.01$, ***$P < 0.001$ vs. IL-1β-treated group. #$P < 0.05$, ##$P < 0.01$, ###$P < 0.001$ vs. control group.

chondrocytes treated with multiple doses of MF extract for 30 h. qRT-PCR analysis revealed that MF extract inhibited *MMP3*, *MMP13*, and *ADAMTS5* expression (Fig 1B). These results were confirmed using western blotting (Fig 4A). Furthermore, as *Col2a1* is a component of the cartilage matrix, its degradation and reduction may initiate and promote OA progression [15]. MF treatment restored the IL-1β-induced decrease in *Col2a1* mRNA levels (Fig 1C). Moreover, western blot analysis revealed that MF treatment increased *COL2A1* expression (Fig 4B). The results were confirmed using immunofluorescence staining (Fig 5). Our findings suggest that the MF extract reduced the levels of OA markers (*MMP3*, *MMP13*, and *ADAMTS5*) and prevented the suppression of *COL2A1* expression.

## Mume Fructus (MF) treatment inhibits LPS-induced inflammation in macrophages

The CCK-8 assay was used to examine the cytotoxic effects of MF on macrophages. The viability of MF- or LPS-treated RAW 264.7 cells was not affected at concentrations of MF ranging from 0 to 100 μg/mL (Fig 6A). LPS can activate macrophages to produce various pro-inflammatory cytokines, such as TNF-α and IL-6, and other inflammatory mediators, including NO [31,32]. To assess the effect of MF on NO production in RAW 264.7 cells, we examined the inflammation induced by LPS and multiple doses of MF extract for over 24 h. MF treatment inhibited NO production in RAW 264.7 cells in a dose-dependent manner with and without LPS stimulation (Fig 6B). Moreover, TNF-α, IL-6, and IL-1β levels significantly increased in LPS-treated cells and decreased in MF-treated cells (Fig 6C).

**Table 4. Chemical profiles of the aqueous extracts of Mume Fructus (MF) and *Prunus mume* (PM).**

| No | Type of phytochemical | Name | Formula | Aqueous extract |
|----|----|----|----|----|
| 1 | Phenylpropanoids | Neochlorogenic acid | $C_{16}H_{18}O_9$ | + |
| 2 | | Chlorogenic acid | $C_{16}H_{18}O_9$ | + |
| 3 | | Cryptochlorogenic acid | $C_{16}H_{18}O_9$ | + |
| 4 | | 1,3-Di-caffeoylquinic acid | $C_{25}H_{24}O_{12}$ | - |
| 5 | | 3,5-Di-caffeoylquinic acid | $C_{25}H_{24}O_{12}$ | - |
| 6 | | 3,4-Di-caffeoylquinic acid | $C_{25}H_{24}O_{12}$ | - |
| 7 | | 3,5-Di-caffeoylquinic acid methyl ether | $C_{26}H_{26}O_{12}$ | - |
| 8 | | Caffeic acid | $C_9H_8O_4$ | - |
| 9 | | cis-*p*-Coumaric acid | $C_9H_8O_3$ | - |
| 10 | | trans-*p*-Coumaric acid | $C_9H_8O_3$ | - |
| 11 | | *p*-Coumaric acid | $C_9H_8O_3$ | - |
| 12 | | Ferulic acid | $C_{10}H_{10}O_4$ | - |
| 13 | | Ferulic acid—cis/trans mixture | $C_{10}H_{10}O_4$ | - |
| 14 | Coumarins | Scopolin | $C_{16}H_{18}O_9$ | - |
| 15 | | Scopoletin | $C_{10}H_8O_4$ | - |
| 16 | Flavonoids | Rutin | $C_{27}H_{30}O_{16}$ | - |
| 17 | | Isoquercetin | $C_{21}H_{20}O_{12}$ | - |
| 18 | | Apigenin-7-glucoside | $C_{21}H_{20}O_{10}$ | - |
| 19 | | Neoliquiritin | $C_{21}H_{22}O_9$ | - |
| 20 | | Catechin | $C_{15}H_{14}O_6$ | - |
| 21 | Phenols | Gallic acid | $C_7H_6O_5$ | - |
| 22 | | 4-Hydroxybenzoic acid | $C_7H_6O_3$ | - |
| 23 | | 2,6-Dihydroxybenzoic acid | $C_7H_6O_4$ | - |
| 24 | Sugar | 5-Hydroxymethyl-2-furaldehyde | $C_6H_6O_3$ | - |

## Mume Fructus (MF) treatment inhibits IL-1β-induced inflammatory signaling in rat chondrocytes

Previous studies have reported that IL-1β-induced MAPK and NF-kB activation promotes MMP overproduction, thereby causing cartilage degradation [12,17]. To confirm the mechanism underlying the effects of MF on rat chondrocytes, the MAPK signaling pathways were assessed using western blot analysis. The p-ERK, p-JNK, and p-p38 levels were significantly increased in IL-1β-stimulated cells and were decreased in MF-treated cells (Fig 7A). Moreover, IL-1β-activated p-p65 expression was inhibited by MF treatment (Fig 7B). Additionally, IL-1β induced the production of various pro-inflammatory cytokines and other inflammatory mediators, including COX-2. MF treatment also inhibited COX-2 (Fig 7C), TNF-α, IL-6, and IL-1β (Fig 7D) expression in rat articular chondrocytes. These results suggest that MF suppressed OA inflammation through MAPK and NF-kB signaling, consequently suppressing the expression of OA markers (*MMP3*, *MMP13*, and *ADAMTS5*) and restoring that of regeneration markers (such as *COL2A1*).

## Discussion

OA is a common joint disease with a steadily increasing prevalence that leads to chronic pain and impaired joint function and poses a global social burden [33,34]. OA is a form of degenerative arthritis that affects the joints of the knees, hips, shoulders, hands, and feet and induces articular cartilage destruction and thinning [1]. Non-surgical modalities for the treatment of

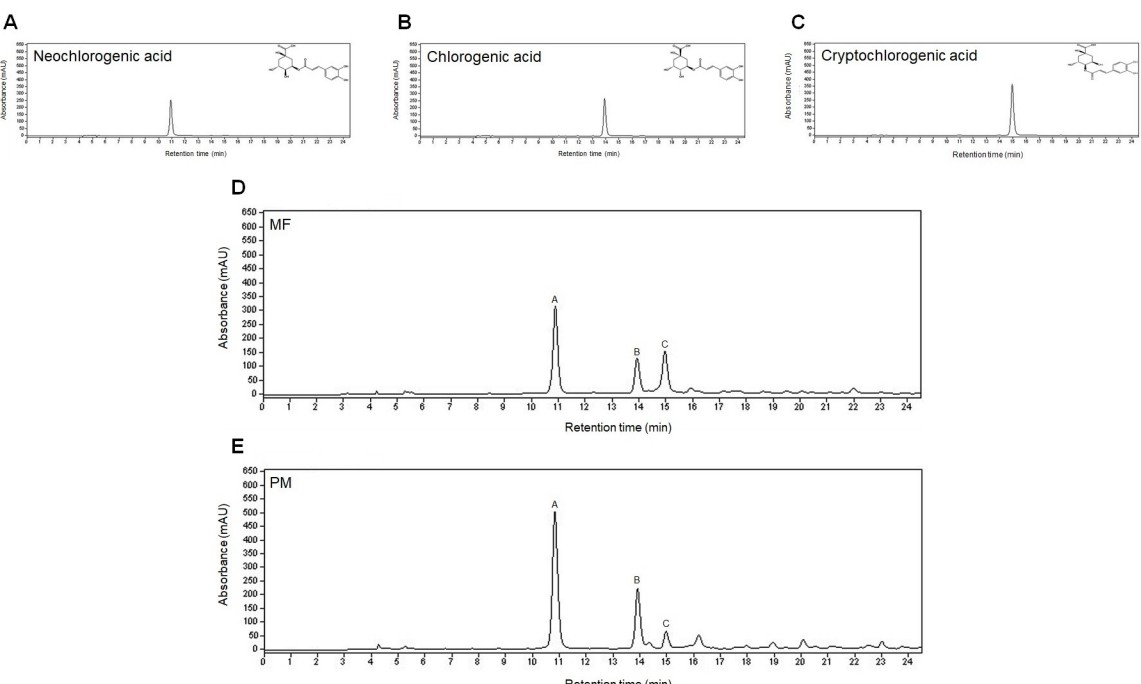

**Fig 3.** Chemical structure of the three main ingredients of Mume Fructus (MF) and *Prunus mume* (PM) extracts, (A) neochlorogenic acid (B) chlorogenic acid, and (C) cryptochlorogenic acid. HPLC chromatograms of (D) MF, and (E) PM extracts. mAU refers to the absorbance unit.

OA include medications such as hyaluronic acid, nonsteroidal anti-inflammatory drugs, and acetaminophen, while surgical treatment options include total knee arthroplasty, osteotomy, and unicompartmental knee arthroplasty [35]. However, complications, such as blood pressure problems and infections, have been reported following some of these treatments [36]. Therefore, plant extracts with joint protective effects and few side effects have been prioritized in the search for an alternative treatment for OA [37–39]. One candidate is MF, a traditional herbal product with anti-inflammatory, antioxidant, and abdominal pain-reducing effects [21,26]; however, its effect on OA remains unknown. PM, a herbal medicinal plant commonly used in traditional Korean medicine and folk remedies, is the fruit of the Chinese plum (*Prunus mume* Sieb. et Zucc.). It has been reported to exhibit antimicrobial activity [40], and is effective against gastric secretion in rats [41] and diabetes [42]. PM has different names and uses depending on the time of harvest and processing method, and it is generally classified as follows: Cheongmae is a green fruit with hard pulp and a strong sour and astringent taste; Cheongmae in steamed and dried state is called Geummae; Cheongmae pickled in *brine* and *sun-dried* is called Baekmae; Cheongmae with its pericarp removed and blackened to charcoal is called Omae (MF); and the yellow fruit with the ripe, fragrant smell is called Hwangmae [43]. MF is processed by removing the pericarp and pits of near-mature Cheongmae picked from mid-June to early July, dried, and steam baked to black in a straw fire. MF is widely used as a medicinal herb in traditional Korean and Chinese medicine [44]. Previous studies on the effects of MF have reported its antioxidant activity [45], antimicrobial effects against bacteria causing food poisoning/gastroenteritis [46], antitumor effects [47], and hypoglycemic effects [48]. However, few studies have specifically investigated the potential of using MF in the treatment of OA.

**Table 5.  Retention times and peak areas of Mume Fructus (MF) and *Prunus mume* (PM) extract (n = 4).**

| Name | | Retention time | Area | Area % | Height | Height % |
|---|---|---|---|---|---|---|
| Neochlorogenic acid | | 10.82 | 3416.25 | 78.56 | 261.84 | 85.00 |
| | | 10.90 | 3347.41 | 78.29 | 256.00 | 85.69 |
| | | 10.96 | 3382.73 | 79.72 | 287.15 | 81.72 |
| | | 10.87 | 3293.04 | 79.31 | 279.02 | 82.90 |
| Chlorogenic acid | | 13.85 | 3699.57 | 76.60 | 274.59 | 79.78 |
| | | 13.90 | 3625.57 | 76.53 | 267.78 | 79.86 |
| | | 13.85 | 3076.21 | 75.58 | 227.75 | 76.33 |
| | | 13.78 | 3558.78 | 76.29 | 263.18 | 76.93 |
| Cryptochlorogenic acid | | 14.93 | 5581.36 | 90.09 | 366.02 | 92.60 |
| | | 14.84 | 5580.47 | 92.84 | 368.36 | 93.38 |
| | | 14.74 | 5854.79 | 90.71 | 384.17 | 92.93 |
| | | 14.63 | 5547.67 | 90.97 | 362.62 | 91.94 |
| Mume Fructus | Neochlorogenic acid | 10.75 | 4194.30 | 37.70 | 248.85 | 34.68 |
| | | 10.76 | 4196.36 | 37.67 | 253.10 | 35.20 |
| | | 10.77 | 4150.94 | 36.93 | 253.46 | 34.93 |
| | | 10.87 | 4041.00 | 37.77 | 313.25 | 39.30 |
| | Chlorogenic acid | 13.82 | 1690.92 | 15.20 | 127.88 | 17.82 |
| | | 13.82 | 1696.46 | 15.23 | 128.58 | 17.88 |
| | | 13.83 | 1697.50 | 15.10 | 128.83 | 17.75 |
| | | 13.91 | 1665.17 | 15.56 | 126.96 | 15.93 |
| | Cryptochlorogenic acid | 14.80 | 2516.18 | 22.62 | 147.30 | 20.53 |
| | | 14.80 | 2524.91 | 22.67 | 148.34 | 20.63 |
| | | 14.81 | 2526.57 | 22.48 | 148.34 | 20.44 |
| | | 14.95 | 2406.64 | 22.50 | 150.63 | 18.90 |
| *Prunus mume* | Neochlorogenic acid | 10.72 | 7363.15 | 39.01 | 530.13 | 37.32 |
| | | 10.72 | 7363.08 | 38.71 | 529.47 | 37.25 |
| | | 10.81 | 7288.23 | 37.15 | 503.99 | 36.23 |
| | | 10.72 | 7381.71 | 39.32 | 527.79 | 37.33 |
| | Chlorogenic acid | 13.83 | 2906.94 | 15.40 | 219.76 | 15.47 |
| | | 13.83 | 2904.89 | 15.27 | 219.57 | 15.45 |
| | | 13.90 | 2851.50 | 14.54 | 217.62 | 15.64 |
| | | 13.83 | 2908.66 | 15.50 | 219.70 | 15.54 |
| | Cryptochlorogenic acid | 14.936 | 1022.46 | 5.42 | 68.90 | 4.85 |
| | | 14.933 | 1022.52 | 5.38 | 69.50 | 4.89 |
| | | 14.955 | 942.33 | 5.02 | 69.72 | 4.94 |
| | | 14.927 | 941.90 | 5.02 | 69.48 | 4.91 |

Area, area under the peak; Area %; percentage of the peak area in the entire chromatogram; Height, peak height; Height %, percentage of the peak height in the entire chromatogram.

PM is used to treat various symptoms, including fatigue, cough, headaches, constipation, food poisoning, and stomach disorders [49]. Pharmacologically, PM has been studied as an antidiabetic [50], hepatoprotective [51], antitumour [52], anti-inflammatory [53,54], antibacterial [55], and anticancer agent [50]. We predicted that PM may also have an anti-inflammatory effect, thereby exerting an inhibitory effect on cartilage destruction due to an inflammatory response. However, PM, the raw material of MF, did not affect OA signaling in the chondrocytes. Assuming that the differential responses could be ascribed to different

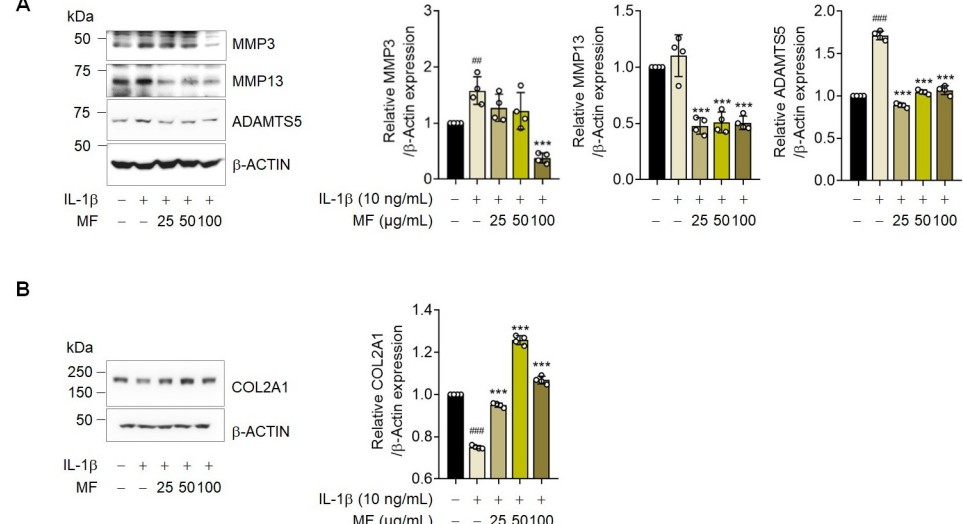

**Fig 4. Effect of Mume Fructus (MF) on the protein levels of metalloproteinase-3 (MMP3), metalloproteinase-13 (MMP13), a disintegrin and metalloproteinase with thrombospondin motifs 5 (ADAMTS5), and collagen type II alpha 1 chain (COL2A1) in rat articular chondrocytes.** Rat articular chondrocytes were treated with or without MF extract (25, 50, and 100 μg/mL) and exposed to IL-1β (10 ng/mL) for 30 h. (A) MMP3, MMP13, ADAMTS5, and (B) COL2A1 protein expression was assessed using western blot analysis (n = 4). Data are expressed as mean ± standard deviation (SD). One-way ANOVA was performed followed by Dunnett's multiple comparisons test. *P < 0.05, **P < 0.01, ***P < 0.001 vs. IL-1β-treated group. #P < 0.05, ##P < 0.01, ###P < 0.001 vs. control group.

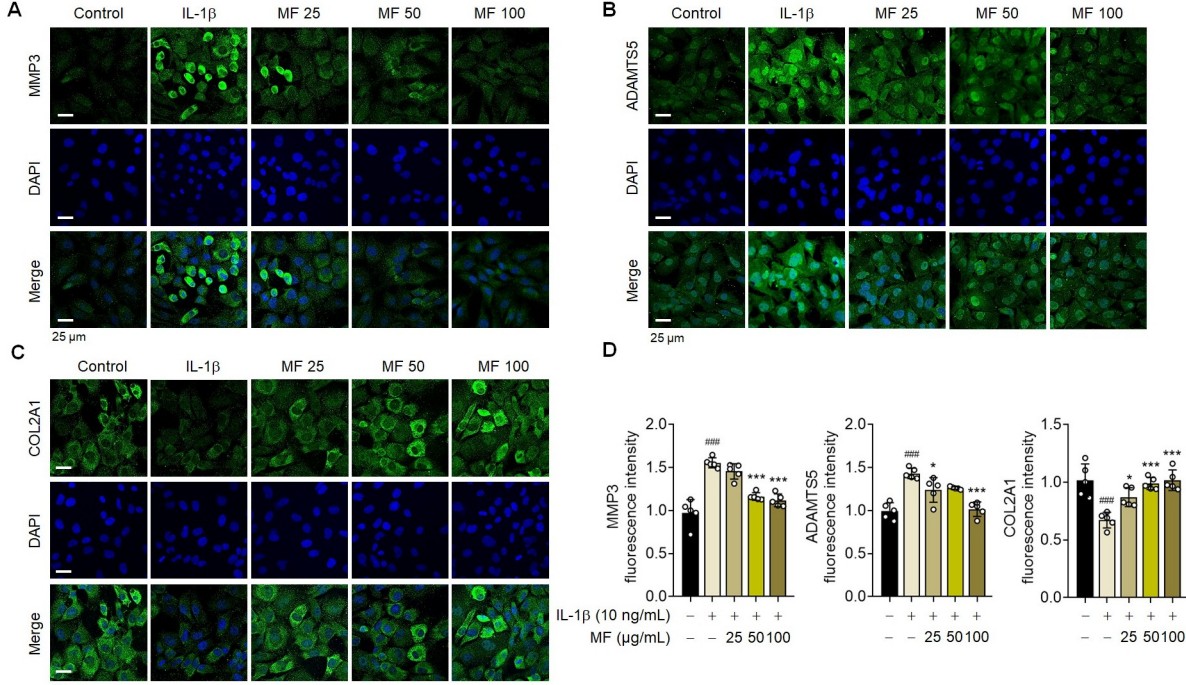

**Fig 5. Effects of Mume Fructus (MF) on IL-1β-induced signaling in primary chondrocytes.** The expression of matrix metalloproteinase (MMP) 3 (A), a disintegrin and metalloproteinase with thrombospondin motifs 5 (ADAMTS5) (B), and collagen type II alpha 1 chain (COL2A1) (C) was evaluated using immunofluorescence staining (n = 5). Scale bar = 25 μm. (D) Fluorescence intensity was quantified using Image J. Data are expressed as mean ± standard deviation (SD). One-way ANOVA was performed, followed by Dunnett's multiple comparisons test. *P < 0.05, **P < 0.01, ***P < 0.001 vs. IL-1β-treated group. #P < 0.05, ##P < 0.01, ###P < 0.001 vs. control group.

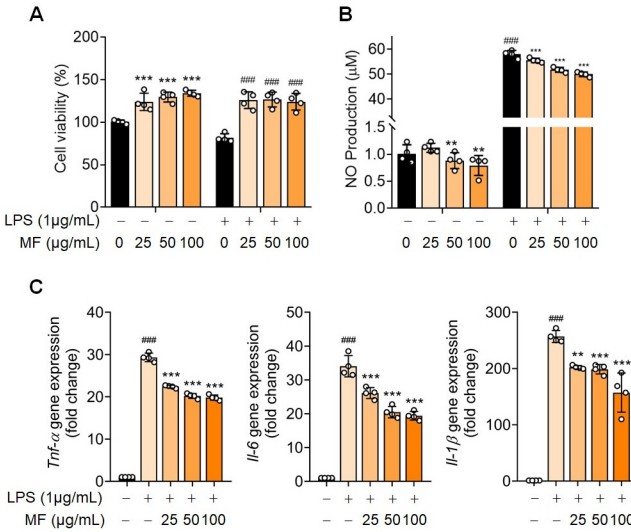

**Fig 6. Effects of Mume Fructus (MF) on lipopolysaccharide (LPS)-induced inflammatory responses in RAW 264.7 cells.** RAW 264.7 cells were treated with or without MF extract (25, 50, and 100 μg/mL) and exposed to LPS (1 μg/mL) for 24 h. (A) Cell viability was assessed using the CCK-8 assay (n = 4). (B) Nitric oxide (NO) production was determined using an NO assay (n = 4). (C) mRNA levels of tumor necrosis factor-alpha (TNF-α), interleukin 6 (IL-6), and interleukin 1 beta (IL-1β) were detected via quantitative reverse transcription-PCR (n = 4). Data are expressed as mean ± standard deviation (SD). One-way ANOVA was performed followed by Dunnett's multiple comparison test. *$P < 0.05$, **$P < 0.01$, ***$P < 0.001$ vs. LPS-treated group. #$P < 0.05$, ##$P < 0.01$, ###$P < 0.001$ vs. control group.

chemical compositions, we conducted phytochemical analysis of MF and PM via HPLC. Our results revealed the presence of three main compounds, with the 4-CQA content differing between MF and PM. 4-CQA is a phenolic acid and unique isomer of chlorogenic acid [56]. Its pharmacological effects include improvement in the blood glucose levels, iron content, and lipid peroxide accumulation via activation of the system XC/glutathione peroxidase 4/factor-erythroid factor 2 system in patients with diabetes [57]. Additionally, 4-CQA attenuates LPS-induced inflammatory responses and oxidative stress and ameliorates myocardial hypertrophy through a hypoxia-inducible factor 1 subunit alpha-related pathway [57,58]. The anti-inflammatory effects of 4-CQA may be attributed to its ability to inhibit OA through MF-mediated inflammatory signaling regulation.

IL-1β induces the expression of several inflammatory mediators, such as IL-6, TNF-α, leukaemia inhibitory factor IL-6 family cytokine, prostaglandin E2, NO, COX-2, and inducible NO synthase, and various catabolic processes; it further contributes to synovial inflammation that degrades the cartilage. This process of IL-1β-induced cartilage destruction activates all three MAPKs (ERK, p38, and JNK) and the NF-κB signal cascade pathway [19]. These intracellular signaling pathways induce OA owing to their participation in ECM degradation homeostasis, resulting in increased *MMP3*, *MMP13*, and *ADAMTS5* levels and reduced *COL2A1* levels [59,60].

The pathophysiology of OA is characterized by the breakdown of the ECM of articular cartilage by proteinases, whose expression is upregulated by inflammatory stimuli [61]. These inflammatory responses occur in chondrocytes and synovial macrophages [62]. Among the pathways involved in inflammatory responses, MAPKs and NF-κB are well known to play key roles [63]. Therefore, we examined and confirmed the anti-inflammatory effects of MF

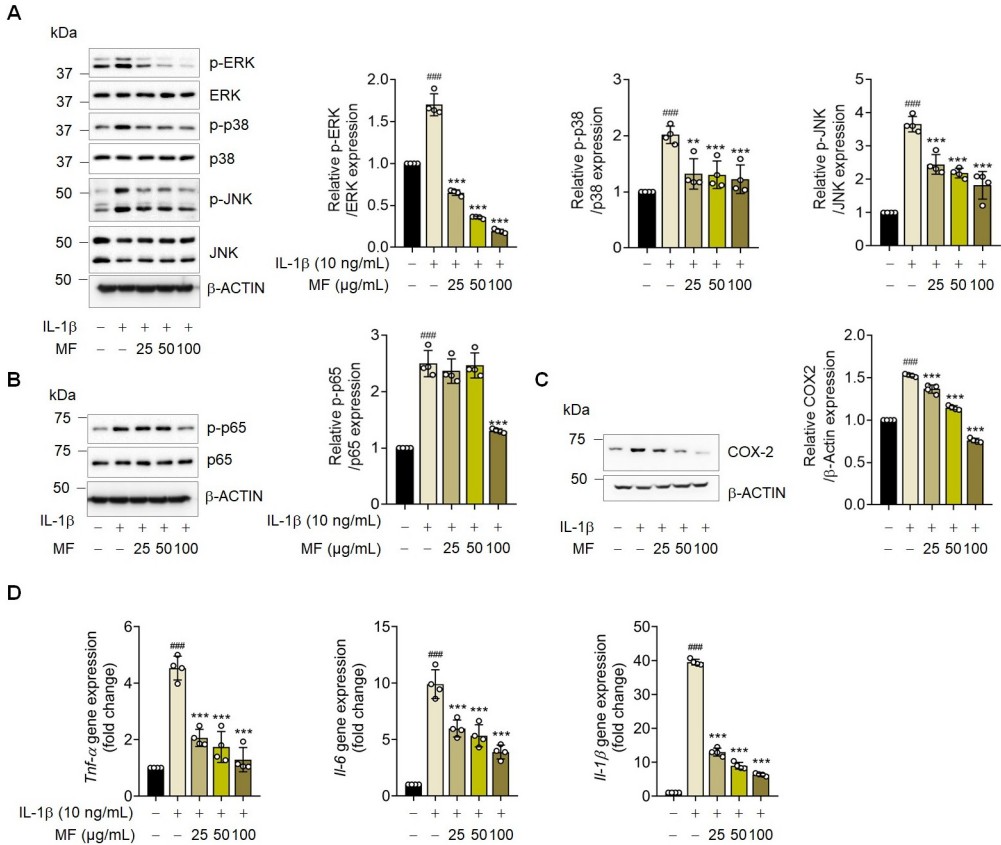

**Fig 7. Inhibition of inflammatory response upon Mume Fructus (MF) treatment via mitogen-activated protein kinase–nuclear factor-kappa B (MAPK–NF-κB) signaling in rat articular chondrocytes.** Rat articular chondrocytes were treated with or without MF extract (25, 50, and 100 μg/mL) and exposed to IL-1β (10 ng/mL) for (A) 30 min, (B) 1 h, and (C, D) 30 h (n = 4). (A) Protein expression of phohphorylated extracellular signal-regulated kinase (p-ERK), phohphorylated c-Jun N-terminal kinase (p-JNK), and phohphorylated P38 (p-p38) assessed via western blot analysis. (B) Protein expression of phohphorylated P65 (p-p65) assessed via western blot analysis. (C) Protein expression of cyclooxygenase-2 (COX-2) assessed via western blot analysis. (D) tumor necrosis factor-alpha (TNF-α), interleukin 6 (IL-6), and interleukin 1 beta (IL-1β) mRNA levels were detected using quantitative reverse transcription-PCR. Data are expressed as mean ± standard deviation (SD). One-way ANOVA was performed followed by Dunnett's multiple comparisons test. *P < 0.05, **P < 0.01, ***P < 0.001 vs. IL-1β-treated group. #P < 0.05, ##P < 0.01, ###P < 0.001 vs. control group.

through gene expression of TNF-α, IL-6, and IL-1β in RAW264.7 cells and also in chondrocytes.

Our results verified the effects of MF treatment in inhibiting MMP3/13 and increasing COL2A1 expression, which were combined to demonstrate the efficacy of MF treatment in OA signaling inhibition (Fig 8). Additionally, MF treatment reduced TNF-α, IL-6, and IL-1β levels in a dose-dependent manner in macrophages and chondrocytes. These OA signaling inhibitions and anti-inflammatory effects were induced by MF, which regulated all three MAPKs and NF-kB signaling. Moreover, MF treatment reduced the levels of COX-2, which is another inflammatory mediator.

Notably, the increase in the COL2A1 expression in the MF treatment group was considerably higher than that in the control group. Collagen is the most abundant protein in mammals [64]. In humans, different types of collagens are present in the connective tissues such as discs,

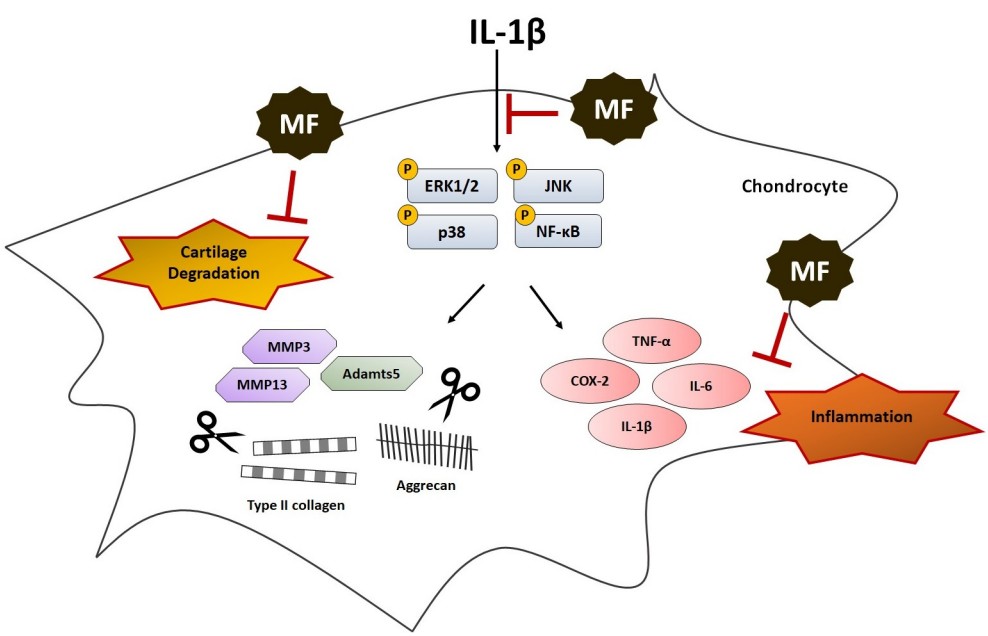

**Fig 8. Mechanistic overview of promoting extracellular matrix regeneration and inhibiting degradation in IL-1β-induced chondrocytes by Mume Fructus (MF).** MF effectively suppressed MAPK/NF-kB signalling after IL-1β-induced damage by significantly suppressing the inflammatory response and consequently promoting extracellular matrix regeneration and inhibiting degradation.

cartilage, tendons, and ligaments of diverse organs such as bones and joints [65]. COL2A1, which comprises discs and cartilages, is degraded by MMP3 and MMP13 [15]. A recent study indicated that the degradation of COL2A1 and a decrease in its level might initiate and promote OA [15]; moreover, the study suggested that collagen degradation played a central role in the pathogenesis of OA. Therefore, the upregulation of COL2A1 expression induced by MF in cartilage may be an OA treatment strategy. Additionally, MF could induce COL2A1 expression and inhibit MMP3 expression in discs during the treatment of herniated discs using Jaseng Woongayoungsin-hwan; however, further research is warranted in this regard.

Our study has a limitation. As we focused on the efficacy of MF extract in chondrocytes, we were able to analyze the marker components of MF but not to the extent of component-specific effects. Further studies are needed to determine the specific effects of 4-CQA on chondrocytes and OA pathogenesis to determine the specific effects of 4-CQA on chondrocytes and OA pathogenesis.

## Conclusions

The current study revealed that MF treatment reduces OA pathogenesis by inhibiting inflammation via MAPK–NF-κB signaling and induces COL2A1 expression. Thus, MF could be a potential therapeutic agent for OA, promoting ECM regeneration and inhibiting its degradation.

## Supporting information

**S1 Table. Chondrocyte cell viability and chondrogenic gene expression.**
(XLSX)

**S2 Table. Immunofluorescence staining data.**
(XLSX)

**S3 Table. Western blot gel intensity.**
(XLSX)

**S4 Table. RAW 264.7 cell viability, NO assay and inflammation gene expression.**
(XLSX)

**S5 Table. Chondrocyte inflammation gene expression.**
(XLSX)

**S1 File. Western blot gel images.**
(PDF)

## Author Contributions

**Conceptualization:** Doo Ri Park.

**Data curation:** Doo Ri Park, Changhwan Yeo, Jee Eun Yoon.

**Formal analysis:** Doo Ri Park, Changhwan Yeo, Jee Eun Yoon.

**Funding acquisition:** In-Hyuk Ha.

**Investigation:** Doo Ri Park, Bo Ram Choi.

**Methodology:** Doo Ri Park.

**Project administration:** In-Hyuk Ha.

**Supervision:** Yoon Jae Lee, In-Hyuk Ha.

**Validation:** Doo Ri Park, Eun Young Hong.

**Visualization:** Doo Ri Park, Changhwan Yeo, Jee Eun Yoon.

**Writing – original draft:** Doo Ri Park.

**Writing – review & editing:** Doo Ri Park, Bo Ram Choi, Seung Ho Baek, Yoon Jae Lee.

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
