## [Decision Letter · Decision Letter 0]

12 Dec 2023

PONE-D-23-27950Mume Fructus reduces interleukin-1 beta-induced cartilage degradation via MAPK downregulation in rat articular chondrocytesPLOS ONE

Dear Dr. Ha,

Thank you for submitting your manuscript to PLOS ONE. After careful consideration, we feel that it has merit but does not fully meet PLOS ONE’s publication criteria as it currently stands. Therefore, we invite you to submit a revised version of the manuscript that addresses the points raised during the review process.

**ACADEMIC EDITOR: **The authors should address all of the comments raised by the three reviewers (see also attachement) and the academic editor before it might be further evaluated.

We look forward to receiving your revised manuscript.

Kind regards,

Adel Tekari, PhD

Academic Editor

PLOS ONE

 [This work was supported by the Jaseng Medical Foundation, Republic of Korea.].  

7. We note that Figure(s) 2A, B, C, 6A, B and C in your submission contain copyrighted images. All PLOS content is published under the Creative Commons Attribution License (CC BY 4.0), which means that the manuscript, images, and Supporting Information files will be freely available online, and any third party is permitted to access, download, copy, distribute, and use these materials in any way, even commercially, with proper attribution. For more information, see our copyright guidelines: http://journals.plos.org/plosone/s/licenses-and-copyright.

a. You may seek permission from the original copyright holder of Figure(s) 2A, B, C, 6A, B and C to publish the content specifically under the CC BY 4.0 license. 

Additional Editor Comments:

After careful consideration of the manuscript, we decided to recommend a major revision. It is a requirement that the authors should address a point-by-point of all the reviewers (total of three) comments before that the manuscript might be further evaluated and considered for publication.

In particular, the authors should change the conclusions in a way that these are support by the results and conduct all the additional and necessary experiments to support their observation. They should also revise their statistical analysis, confirm the number of replicates (biological and technical) being used and add statistics to western blots and/or immunofluorescence.

We are looking forward for an improved version of the manuscript with the necessary data, revised manuscript and the answers to reviewers.

Good luck.

Reviewers' comments:

Reviewer's Responses to Questions

**Comments to the Author**

1. Is the manuscript technically sound, and do the data support the conclusions?

Reviewer #1: Yes

Reviewer #2: No

Reviewer #3: Yes

2. Has the statistical analysis been performed appropriately and rigorously? 

Reviewer #1: Yes

Reviewer #2: No

Reviewer #3: Yes

3. Have the authors made all data underlying the findings in their manuscript fully available?

Reviewer #1: Yes

Reviewer #2: Yes

Reviewer #3: Yes

4. Is the manuscript presented in an intelligible fashion and written in standard English?

Reviewer #1: Yes

Reviewer #2: No

Reviewer #3: Yes

5. Review Comments to the Author

Reviewer #1: The primary goal of the manuscript titled "Mume Fructus reduces interleukin-1 beta-induced cartilage degradation via MAPK down regulation in rat articular chondrocytes" is to clarify the biological process by which a herbal remedy works to mitigate the damaging effects of IL1β on cartilage, using this well-established in vitro model of osteoarthritis.

To support this theory, the authors have looked into the expression of key ECM-destructive enzymes, inflammatory cytokines, and components of the MAPK signaling cascade. They also looked at how MF affected macrophage inflammation brought on by LPS. The findings indicate that MF treatment reduces inflammation in LPS-treated macrophages and decreases OA signaling in chondrocytes produced by IL-1β.

They suggested that Mume Fructus (MF) could be a promising therapeutic agent for OA based on their hypothesis that MF treatment lowers OA pathogenesis by decreasing inflammation via MAPK–NF-κB signaling.

This assertion is corroborated by the western blotting technique results as well as qPCR analysis. After utilizing HPLC to examine the chemical content of MF, the researchers came to the conclusion that the 4-CQA component of MF is what inhibits OA signaling in chondrocytes.

Overall, the authors' results are well-supported by thorough research, and the manuscript itself has theoretical and structural significance. Nonetheless, the writers must address the following several issues:

1. The authors concluded that MF inhibited interleukin-1 beta-induced cartilage degradation via MAPK down regulation, citing references 17–19. They also mentioned that IL1β induces its destructive effects through activation of the MAPK signaling pathway. More proof, though, is required to support this. This theory might be validated by gene silencing certain MAPK signaling pathway components.

2. What software has been utilized for the analysis of immunofluorescence and western blot images?

3. Fig. 3 (B&C) are the same whether LPS treatment is applied or not. They are combinable.

4. Table 2 should include a description of the abbreviations for the phosphorylated forms of NF-ΚB, p65, ERK, JNK, and P38.

5. Verify the chromogram again in Fig. 7-D. There is no greater abundance of the 4-CQA peak in MF extract than in PM.

6. In Discussion, lines 346–348 "The current study revealed that…… chondrocytes," but we are unable to draw the conclusion that the primary cause of the decrease in MMP3/13-ADAMTS-5 is the accumulation of COL2A as a result of MF treatment of chondrocytes. While over-expression of COL2A1 may be a useful treatment strategy for osteoarthritis, it is unable to decrease matrix metalloproteinase enzyme expression.

7. The authors found that the NFĸB and MAPK pathways are in charge of the OA cartilage destruction after studying the IL1β induced inflammatory signaling pathway in rat chondrocytes. Along these lines, a pertinent question concerning LPS-induced inflammation in macrophages emerges and needs to be addressed. Is there a comparable signaling pathway regarding the inflammation that LPS causes in macrophages?

Reviewer #2: This manuscript explores the effects of MF and PM on normal rat chondrocytes in an inflammatory environment induced by IL1β, which simulates osteoarthritic conditions, along with testing these extracts for the inti-inflammatory properties in RAW 264.7 cells. The authors report a dose dependent reduction of MMP3, MMP13 and ADAMTS4 at gene and protein levels as well as induction of COL2A1 production. In addition, they report decrease of NO and proinflammatory chemokines like TNFα, IL6 and IL1β but only in MF-treated cells due to increased levels of 4-CQA in the PM extract.

The conclusions that the authors report are not supported by the results and a major point is repletion of the results presented in the manuscript which are not consistent. Below are some minor and major points that lead to rejection of the manuscript at this current form.

1. Metallopeptidases should be replaced with metalloproteinases throughout the manuscript.

2. Line 64: Are there any references for the use of Jaseng Woongayoungsin-hwan in herniated discs and lower limb pain?

3. Line 71: improves life functions needs to be rephrased. The same for the next sentence lines 72-74.

4. Since MF and PM are basically the same fruit, why both were tested?

5. GAPDH antibody is missing from Table 2.

6. Why concentrations of MF ranging from 0 to 100 μg/mL were used? What is the evidence that the activity is efficient in this range?

7. According to Fig1A, MF treatment seems to induce cell proliferation. How was this taken into consideration for all the subsequent experiments?

8. Why are there no statistical analysis in western blots? If the authors have not performed a statistical test, all statements for significant results need to be rephrased.

9. No information for the biological/technical replicates has been included in methods or in the legends.

10. The representative images and the quantification in Fig2C are not in agreement.

11. What is the reason for using GAPDH in some WB and ACTβ in other for normalisation?

12. Since the authors aim to show the anti-inflammatory a=effects of MF in RAW 264.7 cells, Western blots for TNFα, IL6 and IL1β must be performed as well.

13. It seems that there is a repetition of results in Fig1 and Fig5 as well as Fig2 and Fig6 with regards to MF. Furthermore, it is not clear what were the concentrations used for MF or PM in Fig6. In addition, since the experiments were repeated, why are the fold changes in all 3 gene expressions different for MF?

14. The authors wrote that “the neochlorogenic and chlorogenic acid contents were similar in MF and PM extracts, but 4-CQA was more abundant in MF than in PM as per peak area value analysis”. Where is the quantification for this? This is not evident from Fig7D and the chromatograms should be presented equally, i.e. all baselines should start at 0 mAU.

15. Finally, the authors conclude that the differences that they observe are due to the higher concentration of 4-CQA in PM. To support this, an isolation of the 3 ingredients should be performed and these should be tested separately in primary cells.

16. A major limitation is that there is no support of these findings from in vivo experiments.

Reviewer #3: Park and coworkers demonstrated a connection between Mume Fructus (MF) and its anti-inflammatory modulation in rat chondrocytes and macrophages. They observed through both RT-PCR and WB analyzes a MF-mediated down-regulation of catabolic genes (MMPs and ADAMTS) and an up-regulation of anabolic genes COL2A1 and aggrecan, which reinforce the concept of MF's anti-inflammatory properties. They also observed similar anti-inflammatory conditions in macrophages, suggesting further confirmation of the benefits of MF in inflammatory modulation. After having observed these benefits, they went into more detail and they observed the role of MF in the regulation of the MAPK-NF-kB inflammatory axis. Last but not least, they detected the molecule responsible for anti-inflammatory benefits (4-CQA) in rat chondrocytes. In my opinion, the paper is well structured and tells a story starting from a general concept and delves further and further into molecular details up to the detection of the molecule associated with the aforementioned anti-inflammatory power. However, the short chapter dedicated to MF/NSAIDs and pleiotropic effects has not been fully developed by the authors. I will attach my comments on this for more information.

6. PLOS authors have the option to publish the peer review history of their article (what does this mean?). If published, this will include your full peer review and any attached files.

Reviewer #1: **Yes: **Davood Nasrabadi

Reviewer #2: No

Reviewer #3: **Yes: **Alessandro Marazza

---

## [Author Response · Author response to Decision Letter 0]

15 Mar 2024

Reviewer #1

1. The authors concluded that MF inhibited interleukin-1 beta-induced cartilage degradation via MAPK down regulation, citing references 17–19. They also mentioned that IL1β induces its destructive effects through activation of the MAPK signaling pathway. More proof, though, is required to support this. This theory might be validated by gene silencing certain MAPK signaling pathway components.

Response: We thank you for your comment. First, we thank the reviewer for the valuable suggestions regarding our experiments on the mechanism involving the MAPK signaling pathways. We conducted the study based on the view that verifying the efficacy of MF in inhibiting IL1β-induced inflammatory markers or cartilage degradation in rat chondrocytes through experiments was the most appropriate method of evaluating the efficacy of MF. The main objective of our research was to explore the function of IL-1β signaling in inflammatory processes as it pertains to our study. Although existing inhibitors might offer insightful data regarding the implicated pathways, we deliberately concentrated on the innate reaction to MF, avoiding the influence of external inhibitors. By adopting this method, we were able to scrutinize the inherent signaling pathways and the natural effects of MF on chondrocytes.

2. What software has been utilized for the analysis of immunofluorescence and western blot images?

Response: We apologize for not providing more detailed information on the analytical methods. We added more description about the method of data analysis under Materials and Methods (Page 9, lines 168-172; Page 10, lines 190-191)

The protein bands were visualized using an Amersham Imager 600 imaging system (GE Healthcare Life Sciences, Uppsala, Sweden) and an enhanced chemiluminescence (ECL) system (Bio-Rad, Hercules, CA, USA). The protein levels were quantified using ImageJ (NIH, Bethesda, Maryland, USA).

The fluorescence intensity was analyzed using ImageJ (NIH, Bethesda, Maryland, USA).

3. Fig. 3 (B&C) are the same whether LPS treatment is applied or not. They are combinable.

Response: We thank you for your comment. Following the suggestion, we revised the manuscript by combining the experimental results in RAW264.7cells in Figure 6. We address this issue in the following text added to the Result (Page 26, lines 513-517). 

To assess the effect of MF on NO production in RAW 264.7 cells, we examined the inflammation induced by LPS and multiple doses of MF extract for over 24 h. MF treatment inhibited NO production in RAW 264.7 cells in a dose-dependent manner with and without LPS stimulation (Fig 6B). Moreover, TNF-�, IL-6, and IL-1β levels significantly increased in LPS-treated cells and decreased in MF-treated cells (Fig 6C).

Fig 6. Effects of Mume Fructus (MF) on lipopolysaccharide (LPS)-induced inflammatory responses in RAW 264.7 cells. RAW 264.7 cells were treated with or without MF extract (25, 50, and 100 μg/mL) and exposed to LPS (1 μg/mL) for 24 h. (A) Cell viability was assessed using the CCK-8 assay (n=4). (B) Nitric oxide (NO) production was determined using an NO assay (n=4). (C) mRNA levels of tumor necrosis factor-alpha (TNF-�), interleukin 6 (IL-6), and interleukin 1 beta (IL-1�) were detected via quantitative reverse transcription-PCR (n=4). Data are expressed as mean ± standard deviation (SD). One-way ANOVA was performed followed by Dunnett’s multiple comparison test. *P < 0.05, **P < 0.01, ***P < 0.001 vs. LPS-treated group. #P < 0.05, ##P < 0.01, ###P < 0.001 vs. control group.

4. Table 2 should include a description of the abbreviations for the phosphorylated forms of NF-ΚB, p65, ERK, JNK, and P38.

Response: We thank you for your comment. We have made the necessary revisions and added the relevant sentence. We address this issue in the following text added to the Materials and methods (Page 10, lines 174-179).

MMP3, metalloproteinase-3; MMP13, metalloproteinase-13; ADAMTS5, a disintegrin and metalloproteinase with thrombospondin motifs 5; COL2A1, collagen type II alpha 1 chain; NF�B, nuclear factor kappa B; p-NF�B, phohphorylated nuclear factor kappa B; ERK, extracellular signal-regulated kinase; p-ERK, phohphorylated extracellular signal-regulated kinase; JNK, c-Jun N-terminal kinase; p-JNK, phohphorylated c-Jun N-terminal kinase; p-P38, phohphorylated P38; COX-2, cyclooxygenase-2

5. Verify the chromogram again in Fig. 7-D. There is no greater abundance of the 4-CQA peak in MF extract than in PM.

Response: We apologize for causing confusion as the description of the results and graphical illustration were not in agreement due to the mix-up in the notations of Mume Fructus (MF) and Prunus mume (PM) in the existing HPLC chromatograms. We now present the newly revised HPLC chromatograms in Fig 3 and detailed information on the retention times, area, etc. for HPLC peaks of Mume fructus (MF) and Prunus mume (PM) extracts in Table 5. We address this issue in the following text added to the Results (Page 23, lines 455-470).

HPLC chromatograms showed peaks of neochlorogenic acid (Fig 3A), chlorogenic acid (Fig 3B), and cryptochlorogenic acid (4-CQA, Fig 3C) in both MF and PM extracts. Unlike the peak area distribution of the PM extract, the peak area of 4-CQA in the MF extract was more than 4 times larger than that in the PM extract; the peak area ratio in the MF extract was 22.62% whereas that in the PM extract was 5.42%, indicating a marked difference between the two peak areas. These results suggest that the 4-CQA component in MF inhibits OA signaling in the primary chondrocytes of rats. These results suggest that the 4-CQA component in MF inhibits OA signaling in the primary chondrocytes of rats.

Fig 3. Chemical structure of the three main ingredients of Mume Fructus (MF) and Prunus mume (PM) extracts, (A) neochlorogenic acid (B) chlorogenic acid, and (C) cryptochlorogenic acid. HPLC chromatograms of (D) MF, and (E) PM extracts. mAU refers to the absorbance unit.

Table 5 Retention times and area peak of Mume fructus (MF) and Prunus mume (PM) extract (n=4).

6. In Discussion, lines 346–348 "The current study revealed that…… chondrocytes," but we are unable to draw the conclusion that the primary cause of the decrease in MMP3/13-ADAMTS-5 is the accumulation of COL2A as a result of MF treatment of chondrocytes. While over-expression of COL2A1 may be a useful treatment strategy for osteoarthritis, it is unable to decrease matrix metalloproteinase enzyme expression.

Response: We apologize for not providing sufficient description and causing difficulties in the reviewer’s understanding. We revised the sentence for a more straightforward understanding of the efficacy of MF. We address this issue in the following text added to the Discussion (Page 31, lines 617-619).

Our results verified the effects of MF treatment in inhibiting MMP3/13 and increasing COL2A1 expression, which were combined to demonstrate the efficacy of MF treatment in OA signaling inhibition (Fig 8).

7. The authors found that the NFĸB and MAPK pathways are in charge of the OA cartilage destruction after studying the IL1β induced inflammatory signaling pathway in rat chondrocytes. Along these lines, a pertinent question concerning LPS-induced inflammation in macrophages emerges and needs to be addressed. Is there a comparable signaling pathway regarding the inflammation that LPS causes in macrophages?

Response: We apologize for not providing sufficient description and causing difficulties in the reviewer’s understanding. We provided additional explanations regarding OA and inflammatory responses. We address this issue in the following text added to the Discussion (Page 30, lines 610-616).

The pathophysiology of OA is characterized by the breakdown of the ECM of articular cartilage by proteinases, whose expression is upregulated by inflammatory stimuli [61]. These inflammatory responses occur in chondrocytes and synovial macrophages [62]. Among the pathways involved in inflammatory responses, MAPKs and NF-кB are well known to play key roles [63]. Therefore, we examined and confirmed the anti-inflammatory effects of MF through gene expression of TNF-�, IL-6, and IL-1� in RAW264.7 cells and also in chondrocytes.

Reviewer #2:

1. Metallopeptidases should be replaced with metalloproteinases throughout the manuscript.

Response: We thank you for your comment. We have made the necessary revision under figure legend (Page 22, lines 440).

metalloproteinase-3 (MMP3)

2. Line 64: Are there any references for the use of Jaseng Woongayoungsin-hwan in herniated discs and lower limb pain?

Response: We thank you for your comment. We added more detailed information from the existing literature on the use of Jaseng Woongayoungsin-hwan. We address this issue in the following text added to the Introduction (Page 4, lines 75-82).

Studies have shown that Woongayoungsin-hwan reduces pain and improves life functions [27, 28]. For patients with a herniated intervertebral disc, herbal medicine treatment with Woongayoungsin-hwan has been proven effective in reducing low back pain, showing an improvement with a quick return to the activities of daily life activities and routine [27]. For patients with muscular atrophy in the lower limbs caused by a condition of peripheral neuropathy, Korean medicine treatment with Woongayoungsin-hwan has exhibited a significant therapeutic effect with a gradual increase in the duration of self-walking exercise time [28].

3. Line 71: improves life functions needs to be rephrased. The same for the next sentence lines 72-74.

Response: We thank you for your comment. We deleted the sentence including “improves life functions” and added more information from the existing literature on the use of Jaseng Woongayoungsin-hwan. We address this issue in the following text added to the Introduction (Page 4, lines 75-82).

Studies have shown that Woongayoungsin-hwan reduces pain and improves life functions [27, 28]. For patients with a herniated intervertebral disc, herbal medicine treatment with Woongayoungsin-hwan has been proven effective in reducing low back pain, showing an improvement with a quick return to the activities of daily life activities and routine [27]. For patients with muscular atrophy in the lower limbs caused by a condition of peripheral neuropathy, Korean medicine treatment with Woongayoungsin-hwan has exhibited a significant therapeutic effect with a gradual increase in the duration of self-walking exercise time [28].

4. Since MF and PM are basically the same fruit, why both were tested?

Response: We apologize for not providing sufficient description and causing difficulties in the reviewer’s understanding. MF and PM are both fruits of Chinese plum, but MF is obtained through a fumigation process by harvesting near-mature fruits. We revised with additional information to the text regarding the differences between MF and PM. We address this issue in the following text added to the Discussion (Page 29, lines 568-584).

PM, a herbal medicinal plant commonly used in traditional Korean medicine and folk remedies, is the fruit of the Chinese plum (Prunus mume Sieb. et Zucc.). It has been reported to exhibit antimicrobial activity [40], and is effective against gastric secretion in rats [41] and diabetes [42]. PM has different names and uses depending on the time of harvest and processing method, and it is generally classified as follows: Cheongmae is a green fruit with hard pulp and a strong sour and astringent taste; Cheongmae in steamed and dried state is called Geummae; Cheongmae pickled in brine and sun-dried is called Baekmae; Cheongmae with its pericarp removed and blackened to charcoal is called Omae (MF); and the yellow fruit with the ripe, fragrant smell is called Hwangmae [43]. MF is processed by removing the pericarp and pits of near-mature Cheongmae picked from mid-June to early July, dried, and steam baked to black in a straw fire. MF is widely used as a medicinal herb in traditional Korean and Chinese medicine [44]. Previous studies on the effects of MF have reported its antioxidant activity [45], antimicrobial effects against bacteria causing food poisoning/gastroenteritis [46], antitumor effects [47], and hypoglycemic effects [48]. However, few studies have specifically investigated the potential of using MF in the treatment of OA.

5. GAPDH antibody is missing from Table 2.

Response: We apologize for causing confusion by using two different housekeeping genes. Both ACTβ and GAPDH are commonly used as housekeeping genes, but due to internal circumstances in the authors’ laboratory, a supply of β-actin antibodies was not available, and we had to use GAPDH instead in the experiment. We performed replicate experiments and revised the results with those obtained from using ACTβ as housekeeping genes instead of the previous results with GAPDH.

6. Why concentrations of MF ranging from 0 to 100 μg/mL were used? What is the evidence that the activity is efficient in this range?

Response: We thank you for your comment. In previous studies using MF, concentrations widely ranging from 0.5, 2, 5, 10, 20, 30, 50, 70, 100, 150, 200, to 300ug/ml were used for experiments [1-4]. In addition, in experiments with chondrocytes using natural products, the concentration in the range of 0 - 100 μg/mL was mainly used; thus considering these ranges of concentration, the concentrations of MF in this study ranged from 0 to 100 μg/mL [5-7].

7. According to Fig1A, MF treatment seems to induce cell proliferation. How was this taken into consideration for all the subsequent experiments?

Response: The cell counting Kit-8, the method we used to perform cell viability assay, is a convenient method to detect cytotoxicity under various conditions or concentrations of samples. Cell viability was confirmed after the treatment of samples, and the counts represent the number of healthy cells. The fact that cell death did not occur in chondrocytes even after treatment with MF for more than 24 hours indicates that no toxicity of MF was confirmed in cell-level experiments. Therefore, the confirmed cell viability may be considered as an indicator that incubation with various doses of MF was performed without problems in chondrocytes and RAW264.7 cell. 

8. Why are there no statistical analysis in western blots? If the authors have not performed a statistical test, all statements for significant results need to be rephrased.

Response: We thank you for your comment. For western blot analysis, we rechecked about replicate measurements, and revised the legends of the Figures and the graphs used. We address this issue in the following text added to the Results (Page 25, lines 490-498; Page 28, lines 542-555).

Fig 4. Effect of Mume Fructus (MF) on the protein levels of metalloproteinase-3 (MMP3), metalloproteinase-13 (MMP13), a disintegrin and metalloproteinase with thrombospondin motifs 5 (ADAMTS5), and collagen type II alpha 1 chain (COL2A1) in rat articular chondrocytes. Rat articular chondrocytes were treated with or without MF extract (25, 50, and 100 μg/mL) and exposed to IL-1� (10 ng/mL) for 30 h. (A) MMP3, MMP13, ADAMTS5, and (B) COL2A1 protein expression was assessed using western blot analysis (n=4). Data are expressed as mean ± standard deviation (SD). One-way ANOVA was performed followed by Dunnett’s multiple comparisons test. *P < 0.05, **P < 0.01, ***P < 0.001 vs. IL-1�-treated group. #P < 0.05, ##P < 0.01, ###P < 0.001 vs. control group.

Fig 7. Inhibition of inflammatory response upon Mume Fructus (MF) treatment via mitogen-activated protein kinase–nuclear factor-kappa B (MAPK–NF-�B) signaling in rat articular chondrocytes. Rat articular chondrocytes were treated with or without MF extract (25, 50, and 100 μg/mL) and exposed to IL-1� (10 ng/mL) for (A) 30 min, (B) 1 h, and (C, D) 30 h (n=4). (A) Protein expression of phohphorylated extracellular signal-regulated kinase (p-ERK), phohphorylated c-Jun N-terminal kinase (p-JNK), and phohphorylated P38 (p-p38) assessed via western blot analysis. (B) Protein expression of phohphorylated P65 (p-p65) assessed via western blot analysis. (C) Protein expression of cyclooxygenase-2 (COX-2) assessed via western blot analysis. (D) tumor necrosis factor-a

---

## [Decision Letter · Decision Letter 1]

16 Apr 2024

Mume Fructus reduces interleukin-1 beta-induced cartilage degradation via MAPK downregulation in rat articular chondrocytes

PONE-D-23-27950R1

Dear Dr. Ha,

We’re pleased to inform you that your manuscript has been judged scientifically suitable for publication and will be formally accepted for publication once it meets all outstanding technical requirements.

Kind regards,

Adel Tekari, PhD

Academic Editor

PLOS ONE

Additional Editor Comments (optional):

Congratulations and good continuation in the field.

Reviewers' comments:

Reviewer's Responses to Questions

**Comments to the Author**

1. If the authors have adequately addressed your comments raised in a previous round of review and you feel that this manuscript is now acceptable for publication, you may indicate that here to bypass the “Comments to the Author” section, enter your conflict of interest statement in the “Confidential to Editor” section, and submit your "Accept" recommendation.

Reviewer #1: All comments have been addressed

Reviewer #3: All comments have been addressed

2. Is the manuscript technically sound, and do the data support the conclusions?

Reviewer #1: Yes

Reviewer #3: Yes

3. Has the statistical analysis been performed appropriately and rigorously? 

Reviewer #1: Yes

Reviewer #3: Yes

4. Have the authors made all data underlying the findings in their manuscript fully available?

Reviewer #1: Yes

Reviewer #3: Yes

5. Is the manuscript presented in an intelligible fashion and written in standard English?

Reviewer #1: Yes

Reviewer #3: Yes

6. Review Comments to the Author

Reviewer #1: The manuscript has been substantially improved.‎ The authors have addressed all questions and improved the manuscript. I have no objection to accepting the article.

Reviewer #3: The authors resolved all technical problems concerning western blot analysis (biological replicates) and they show consistency in their data. Most of my questions were not answered, but the authors decided to be more focused in their project and to avoid to answer some of my questions concerning investigation of additional inflammatory pathways. Although it's a pity to not have a wider story concerning Mume Fructus and its potential anti-inflammation benefit, the data showed in this paper is valid and, as already said before, consistent. For this reason, my last word is to accept the article for pubblication.

7. PLOS authors have the option to publish the peer review history of their article (what does this mean?). If published, this will include your full peer review and any attached files.

Reviewer #1: **Yes: **Davood Nasrabadi

Reviewer #3: **Yes: **Alessandro Marazza

---

## [Editor Report · Acceptance letter]

26 Apr 2024

PONE-D-23-27950R1 

PLOS ONE

Dear Dr. Ha, 

I'm pleased to inform you that your manuscript has been deemed suitable for publication in PLOS ONE. Congratulations! Your manuscript is now being handed over to our production team.

Kind regards, 

on behalf of

Dr. Adel Tekari 

Academic Editor

PLOS ONE